# α-Fodrin in Cytoskeletal Organization and the Activity of Certain Key Microtubule Kinesins

**DOI:** 10.3390/genes12050750

**Published:** 2021-05-17

**Authors:** Jamuna S. Sreeja, Athira Jyothy, Suparna Sengupta

**Affiliations:** Cancer Research Program, Rajiv Gandhi Centre for Biotechnology, University of Kerala, Thiruvananthapuram 695014, India; jssreeja@rgcb.res.in (J.S.S.); athirajyothy@rgcb.res.in (A.J.)

**Keywords:** fodrin, spectrin, kinesins, LC-MS, proteomics

## Abstract

Cortical cytoskeletal proteins are significant in controlling various cellular mechanisms such as migration, cell adhesion, intercellular attachment, cellular signaling, exo- and endocytosis and plasma membrane integrity, stability and flexibility. Our earlier studies involving in vitro and ex vivo approaches led us to identify certain undiscovered characteristics of α-fodrin, a prominent cortical protein. The conventional functions attributed to this protein mainly support the plasma membrane. In the present study, we utilized a global protein expression analysis approach to detect underexplored functions of this protein. We report that downregulation of α-fodrin in glioblastoma cells, U-251 MG, results in upregulation of genes affecting the regulation of the cytoskeleton, cell cycle and apoptosis. Interestingly, certain key microtubule kinesins such as KIF23, KIF2B and KIF3C are downregulated upon α-fodrin depletion, as validated by real-time PCR studies.

## 1. Introduction

The cortical cytoskeleton is an important feature contributing towards the integrity and strength of cells. Metazoan cells have evolved various functional proteins to serve as cortical proteins supporting the inner aspect of the plasma membrane. One such family of proteins is the spectrin family. The major representative members of this family are spectrin and fodrin. Both these proteins have been implicated in supporting the plasma membrane. While spectrin is mostly found in the erythrocytes, fodrin is more ubiquitous in presentation, as it is expressed in most tissues apart from the erythrocytes, and most abundantly in brain tissue [1].

Fodrin has been shown to be important in a wide variety of functions such as signal transduction, and cardiac and brain development [2,3]. Through our studies, it was understood that fodrin not only serves as a cortical supporting protein but also interacts with γ-tubulin and inhibits its microtubule nucleation activity [4]. We have also shown that α-fodrin, a subunit of fodrin, is vital to the movement of γ-tubulin to the centrosome and is therefore critical in microtubule nucleation from the centrosome in brain-derived cell lines (unpublished data). The literature survey also revealed diverse functions of this protein. Further, we have been able to show that α-fodrin is important in chromosome congression and mitotic progression in cells [5]. It is also important in calcium-based signaling, especially in neuronal cells [6]. Fodrin has also been implicated in vesicular movement through the axons for neurite building [7]. 

To gain a comprehensive view on the diverse functions of fodrin, it was imperative to look into the downstream effectors of this protein in an unbiased fashion. Hence, we decided to downregulate α-fodrin in glioblastoma cells, U-251 MG, and then perform a global protein expression analysis by LC-MS/MS. We chose to work with glioblastoma cells because of the abundant presence of α-fodrin. Additionally, through our previous studies, we have been able to understand that α-fodrin is significant in microtubule formation (unpublished results) and organization in these cells [4,5,8].

## 2. Materials and Methods

### 2.1. Maintenance of Cell Culture

The U-251 MG (glioblastoma) cell line was procured from Sigma-Aldrich (St. Louis, MO, USA). It was maintained in MEM with non-essential amino acids, 1 mM sodium pyruvate, 10% FBS and 1× antibiotic mixture (100 units/mL of penicillin, 100 units/mL of streptomycin and 0.25 μg/mL of Amphotericin B). The cells were incubated at 37 °C in a CO_2_ incubator maintaining humid conditions and 5% CO_2_. The subculturing and cryopreservation of cells were conducted according to the prescribed guidelines.

### 2.2. Total Protein Extraction for Proteomic Study

U-251 MG cells were treated with control shRNA/α-fodrin shRNA for 96 h to achieve appreciable downregulation of α-fodrin expression. The shRNA used for these experiments has beendetailed in our previous publication [5]. The cells were then harvested and total protein was extracted using RapiGest^TM^ SF (Waters Corporation, Milford, MA, USA). It was added at a final concentration of 0.5% in 50 mM ammonium bicarbonate with protease inhibitor cocktail to the cell pellet and incubated in ice for 45 min with intermittent vortexing. To break open the cells, the samples were sonicated in a bath sonicator for 3 cycles of 10 min each at 37 °C followed by three cycles of freeze–thawing in liquid N_2_. The samples were centrifuged at 14,000 rpm at 4 °C for 15 min, and the supernatants containing crude proteins were retained and estimated.

### 2.3. In-Solution Trypsin Digestion

Approximately 100 µg of total protein from each sample, normalized to a concentration of 1 µg/µL, was subjected to in-solution trypsin digestion to generate peptides. Disulfide bonds were reduced by treatment with 100 mM dithiothreitol in 50 mM ammonium bicarbonate (ABC) for 30 min at 60 °C. After cooling at room temperature for 5 min, the samples were incubated with 200 mM iodoacetamide in 50 mM ABC to perform alkylation in dark at room temperature. Proteins were then digested by using sequencing grade modified trypsin (Sigma) at a trypsin/total protein ratio of ~1:25 and incubated for 17 h at 37 °C. The enzymatic reaction was stopped by adding formic acid to each sample so that the final formic acid concentration was 1.0%, followed by incubation at 37 °C for 20 min. The digested peptide solutions were centrifuged at 20,817× *g* for 12 min, and the supernatant was stored at −20 °C until LC-MS/MS analysis.

## 3. LC-MS

The tryptic peptides were separated using a nanoACQUITY UPLC^®^ chromatographic system (Waters, Manchester, UK), and the instrument control and data processing were conducted with MassLynx4.1 SCN781 software. The peptides were separated by reversed-phase chromatography and injected in partial loop mode in a 5 μL loop (injection volume 3.0 μL). Water and acetonitrile were used as solvent A and B, respectively. All solvents for the UPLC system contained 0.1% formic acid. The peptides were trapped and desalted on a trap column (Symmetry^®^ 180 µm × 20 mm C18 5 µm, Waters) for 1 min at a flow rate of 15 μL/min. The trap column was placed in line with the reversed-phase analytical column, a 75 µm (internal diameter) x 200 mm HSS T3 C18 (Waters) with particle size of 1.8 µm. Peptides were eluted from the analytical column with a linear gradient of 1 to 40% solvent B over 55.5 min at a flow rate of 300 nL/min, followed by a 7.5 min rinse of 80% solvent B. The column was immediately re-equilibrated at initial conditions (1% solvent B) for 20 min. The column temperature was maintained at 40 °C. The lock mass, [Glu1]-Fibrinopeptide B human (Sigma) (positive ion mode [M+2H]^2+^ = 785.8426), for mass correction was delivered from the auxiliary pump of the UPLC system through the reference sprayer of the NanoLockSpray^TM^ source at a flow rate of 500 nL/min. Each sample was injected in triplicate with blank injections between each sample.

MS analysis of eluting peptides was carried out on a SYNAPT^®^ G2 High Definition MS™ System (HDMSE System (Waters). The instrument settings were: nano-ESI capillary voltage—3.4 KV, sample cone—40 V, extraction cone—4 V, IMS gas (N2) flow—90 (mL/min). To perform the mobility separation, the IMS T-Wave™ pulse height was set to 40 V during transmission and the IMS T-Wave™ velocity was set to 800 m/s. The travelling wave height was ramped over 100% of the IMS cycle between 8 and 20 V.

All analyses were performed in positive mode ESI using a NanoLockSpray^TM^ source. The lock mass channel was sampled every 45 s. The time-of-flight analyzer (TOF) of the mass spectrometer was calibrated with a solution of 500 fmole/μL of [Glu1]-Fibrinopeptide B human (Sigma). This calibration set the analyzer to detect ions in the range of 50–2000 m/z. The mass spectrometer was operated in resolution mode (V mode) with a resolving power of 18,000 FWHM, and the data acquisition was performed in continuum format. The data were acquired by rapidly alternating between two functions—Function-1 (low energy) and Function-2 (high energy). In Function-1, only low-energy mass spectra (MS) were acquired, and in Function-2, mass spectra at elevated collision energy with ion mobility (HDMSE) were acquired. In Function-2, collision energy was set to 4 eV in the trap region of the mass spectrometer and was ramped from 20 to 45 eV in the transfer region of the mass spectrometer to attain fragmentation in the HDMSE mode. The continuum spectral acquisition time in each function was 0.9 s with an interscan delay of 0.024 s.

## 4. MS Data Analysis

The acquired ion mobility enhanced MSE spectra were analyzed using Progenesis QI for Proteomics V3.0 (Non Linear Dynamics, Waters) for protein identification as well as for the label-free relative protein quantification. Data processing included lock mass correction post-acquisition. Processing parameters for Progenesis were set as follows: noise reduction thresholds for low-energy scan ion—150 counts, high-energy scan ion—30 counts. The protein identifications were obtained by searching against the human protein database downloaded from UniProt. During the database search, the protein false positive rate was set to 4%. The parameters for protein identification were made in such a way that a peptide was required to have at least 1 fragment ion match, and a protein was required to have at least 3 fragment ion matches and at least 1 peptide match for identification. Oxidation of methionine was selected as a variable modification and cysteine carbamidomethylation was selected as a fixed modification. Trypsin was chosen with a specificity of 1 missed cleavage. Datasets were analyzed and quantified by relative quantitation using the Hi-N algorithm, which resolves peptide conflicts and uses the average intensity of the three most abundant unique peptides for a protein. Furthermore, only a fold change higher than 50% difference (ratio of either <0.50 or >1.50) was considered to be indicative of significantly altered levels of expression. The analysis was conducted using ANOVA and a *p*-value of <0.05 was considered to be significant.

### 4.1. Identification of Commonly Affected Proteins

After applying the cut-offs as stated above and filtering the hits, a sorted list of proteins was obtained. The experiment was performed in triplicates (three biological and three technical repeats) and result lists finally obtained were submitted into the VENNY 2.1 (https://bioinfogp.cnb.csic.es/tools/venny/, accessed on 5 April 2019) online tool to obtain a list of commonly affected proteins. A total of 2045 proteins were obtained. These proteins were filtered through the following cut-offs: peptide count of more than 2, ANOVA score of less than 0.05 and a fold change of more than 1.5. A total of 247 proteins passed these criteria, from which 78 showed higher abundance and 169 showed reduced abundance in α-fodrin shRNA cells compared to control shRNA-treated cells. The obtained protein list was submitted into the UNIPROT database to obtain the individual IDs and other related information about these proteins.

### 4.2. Bioinformatics Analysis of the Affected Proteins

The identified proteins were subjected to Gene Ontology (GO) analysis by Annotation, Visualization and Integrated Discovery (DAVID) functional Annotation Bioinformatic analyzer v 6.7. The pathways obtained were from KEGG (Kyoto Encyclopedia of Gene and Genomes), PANTHER and REACTOME. DAVID generates an Expression Analysis Systematic Explorer (EASE) score, a modified Fisher’s exact *p*-value, for each term. The GO terms with a *p*-value ≤ 0.1 were considered as enriched. Protein–protein interaction networks of differentially overexpressed proteins were generated using STRING v 11.0 (https://string-db.org/, accessed on 6 April 2019) and a confidence view was generated by setting the filter to high confidence (0.700).

### 4.3. Quantitative Real-Time PCR

U-251 MG cells were transfected either with control shRNA or α-fodrin shRNA. At 96 h post-transfection, RNA was isolated from the samples using Qiagen RNeasy mini kit. The purified RNA was used as template for first-strand cDNA synthesis using Verso cDNA synthesis kit. SYBR Green premix reagent from TAKARA was used to quantify the cDNA using the following gene-specific primers (Table 1). The relative changes in gene expression between control shRNA- and α-fodrin shRNA-treated cells were calculated using the 2^−ΔCt^ method. β-Actin was used as an internal expression control because we had earlier understood and reported that the expression level of β-Actin does not alter upon α-fodrin downregulation [5]. Experiments were conducted in duplicates and the results were statistically analyzed using standard deviation and standard error of mean.

### 4.4. Western Blotting

Cell lysates from U-251 MG control shRNA- and α-fodrin shRNA-treated cells were extracted using phospholysis buffer. The lysates were run on 10% SDS-PAGE. The protein bands were transferred onto a pre-activated PVDF membrane and probed with anti-KIF 2B/anti-KIF 3C/anti-KIF 23 rabbit antibodies. HRP-conjugated anti-rabbit secondary antibody was used to develop the protein bands corresponding to individual proteins.

## 5. Results 

### Protein Identification

To understand the downstream effectors of α-fodrin downregulation in U-251 MG cells, α-fodrin was specifically downregulated using targeted shRNA transfection. The protein lysates from control and α-fodrin shRNA-treated cells were subjected to in-solution trypsin digestion followed by LC-MS/MS. The experiment was conducted in triplicates. The hits (2045 proteins) obtained from all three sets were subjected to VENNY 2.1 to obtain a final list of common proteins (Figure 1A). The proteins with at least 2 peptide counts, an ANOVA score of <0.05 and a maximum fold change of ≥1.5 were selected for further analysis. A total of 247 proteins passed all the cut-off criteria and amongst these, 78 showed higher abundance and 169 showed reduced abundance in α-fodrin shRNA-transfected cells compared to control shRNA-treated cells (Figure 1B). Various protein databases were used to analyze these proteins. Since cellular pathways are highly interconnected and do not work in isolation, there is considerable overlap and hence different proteins can occur in multiple pathways, which accounts for the redundancy in the presented data. However, it should be noted that this does not affect our interpretation because only significantly enriched clusters have been detailed. In the present study, we have also discussed, in detail, a few interesting proteins in the upregulated and the downregulated cohort. These proteins are of general interest and also overlap with the research focus of our laboratory.

## 6. Analysis of the Upregulated Proteins

For the initial analysis of the 78 upregulated proteins, their accession IDs, obtained from the proteomics results, were submitted to UNIPROT. The protein names and corresponding gene names were thus identified (Appendix A). We then analyzed the interactome of these upregulated proteins through STRING v11.0. (Figure 2A).

The STRING analysis revealed the presence of a few interesting clusters, the cytoskeletal regulatory proteins, the cell cycle proteins and the apoptotic proteins. We further conducted a Gene Ontology analysis of the upregulated proteins using DAVID v 6.7 (Figure 2B). The pathways from KEGG, PANTHER and REACTOME databases were obtained. All hits had a *p*-value of <0.05, and an EASE score (a modified Fischer exact) value of <0.1 was used as the cut-off criterion. The data obtained corroborated the STRING analysis. The cytoskeletal protein cluster obtained in the STRING output also showed up in the pathway analysis (Figure 3A, marked with * in the graph). Biological processes such as gene expression, translation and intracellular transport are also affected by the upregulated proteins (Figure 3B).

## 7. Analysis of the Downregulated Proteins

As conducted for the upregulated proteins, 169 downregulated proteins were identified using the UNIPROT database and the corresponding gene names and other characteristics were obtained (Appendix A). Further, the downregulated proteins were analyzed using STRING v 11.0. The interactome is depicted in Figure 4A. Enriched clusters of gene expression proteins, microtubule cytoskeleton proteins and DNA replication proteins were obtained.

STRING analysis revealed an interesting cluster, the microtubule cytoskeleton cluster. This was of relevance to our study because we have earlier described the regulatory effect of fodrin on microtubule nucleation [4]. We have also understood the importance of α-fodrin in microtubule formation in cells (unpublished data). The downregulated proteins were further used for pathway analysis using DAVID v 6.7. Pathways from KEGG, REACTOME and PANTHER appeared enriched. The cut-off criterion was the same as for upregulated proteins. It was found that several pathways such as the gene expression pathways, the focal adhesion pathways and the DNA replication pathways were downregulated upon α-fodrin downregulation (Figure 4B). The most important proteins that were obtained as leads are kinesin family members KIF2B, KIF3C and KIF23. The literature survey of these kinesins (as will be detailed later) showed their relevance in various processes such as chromosome congression, spindle morphology and cargo transport. These pathways resonated with our understanding of fodrin functions. Hence, it was imperative for us to analyze and validate them. We performed real-time PCR using specific primers and estimated the first-strand cDNA synthesis (Figure 5). Expression analysis of each gene was calculated based on at least two independent experiments. Relative expression values were calculated as 2^−Δ(CTt^^arget–CT^^reference)^ with β-actin as the housekeeping gene. An earlier report published from our laboratory showed that the total actin amount in the cell remained the same upon α-fodrin downregulation [5]. We found that KIF3C and KIF23 mRNA expression was appreciably downregulated in α-fodrin downregulated cells. However, we could not obtain any significant alteration in the levels of KIF2B. The KIF2B mRNA might have a longer half-life due to which we could not appreciate any difference in its mRNA levels (Figure 5A). However, we performed Western blotting on control and α-fodrin-depleted cell lysates and found that all the three kinesins reduced convincingly (Figure 5B).

We further performed an analysis of the GO terms corresponding to the cellular components enriched as a result of the downregulation of α-fodrin. It was revealed that the microtubule cytoskeleton and the spindle are affected (Figure 6A,B), marked by * in the figure. This concorded with the STRING analysis of the data and qPCR analysis of the mRNA expression and Western blot analysis.

## 8. Discussion

Fodrin downregulation caused an array of upregulated and downregulated proteins in U-251 MG cells, as analyzed by a proteomics approach. The proteins that were shown to be upregulated were analyzed by STRING v 11.0 and DAVID v 6.7 to understand the protein interactome and the affected pathways. The STRING analysis revealed the presence of a few interesting clusters, the cytoskeletal regulatory proteins, the cell cycle proteins and the apoptotic proteins. Rho GTPases are small regulatory GTP-binding proteins belonging to the Ras superfamily. Their ability to bind and hydrolyze GTP enables them to act as switches in various signal transduction pathways. These proteins are also regulated at multiple levels by GTP-activating proteins (GAPs), GTP exchanging factors (GEFs) and GDP dissociation inhibitors (GDIs). Rho GTPases affect cytoskeletal regulation in two major modes—regulating the actin stress fiber formation and affecting the microtubule dynamic instability. The activated Rho GTPases induce the formation of the actin–myosin complex which protrudes in the lamellipodia [9]. In the present proteomic study, it was found that α-fodrin downregulation results in upregulation of RhoB and RhoC proteins. Fodrin is a large heterodimeric protein that houses two actin-binding calpain homology domains in its β-subunit. Fodrin’s association with actin is important for the integrity of the plasma membrane. However, it has been reported that actin reorganizes under conditions of stress to form actin bundles [10,11]. Fodrin has also been shown to re-polarize under conditions of stress in specific cells such as chromaffin cells [12]. Hence, it is possible that downregulation of α-fodrin may result in the induction of various signaling pathways that induce actin stress fiber formation by Rho GTPases.

Rho GTPases, upon activation, inhibit microtubule end-binding proteins such as stathmin. This in turn reduces the dynamic instability of microtubules, resulting in stable microtubules, especially at focal adhesion points [9,13]. Independent of this, it was also found that stathmin was downregulated upon α-fodrin depletion (Appendix A). Stathmin is a microtubule depolymerizing protein. Reduction in stathmin via activation of Rho GTPases can thus result in stable microtubules. Fodrin has been shown to associate with tubulin and induce microtubule bundling in vitro [14]. However, there is no direct evidence to validate the effect of fodrin on microtubule plus end dynamics. Thus, through the presented proteomic evidence, it is indicated that absence of α-fodrin may indirectly affect the dynamic instability of microtubules.

In the present expression analysis, it was found that certain cell cycle regulatory proteins such as cyclin-dependent kinases 11a and 11b (CDK11a and CDK11b) were found to be upregulated. Literature analysis of these proteins shows that overexpression of CDK11 influences the cell cycle in CHO cells [15]. Increased presence of CDK11B results in delayed late telophase and inhibits entry into S-phase [16,17]. A direct link of α-fodrin in telophase could not be obtained; however, it is reiterated through multiple reports to be important in cell cycle progression. Reduction in α-fodrin in the human melanoma WM-266 cell line resulted in G1 arrest [18]. Apart from this, interactome analysis of α-fodrin shows the presence of multiple cell cycle regulatory proteins such as CDK2 which is important for the G1–S transition. Another independent study conducted in our lab showed that α-fodrin depletion caused delayed mitotic progression [5]. These facts indicate a formidable role of α-fodrin in the cell cycle regulation; however, detailed experiments would be required to expand the present understanding of this aspect. 

Through our proteomic analysis, we understand that α-fodrin downregulation results in an increase in apoptotic pathway proteins such as 26S proteasome non-ATPase regulatory subunit 14 and polyubiquitin C and B. Ubiquitination and proteosome-based protein degradation are pathways that accompany apoptosis in cells. They are also important regulators of the apoptotic proteins [19]. The literature survey revealed that α-fodrin is also associated with apoptosis. α-fodrin carries a caspase cleavage site in the central portion of the molecule. Such specific cleavage of α-fodrin is associated with the classic membrane blebbing, a signature aspect of apoptosis [20]. However, through our analysis, our proteomic data reveal that a reduced presence of α-fodrin can also result in apoptosis accompanying the ubiquitination and targeted degradation of specific proteins through the 26S proteasome. 

Apart from these, many other pathways such as the gap junction pathway and PDGF, Wnt and EGFR signaling pathways were also upregulated, to name a few. We also looked into the molecular function Gene Ontologies that were enriched by these upregulated proteins (Figure 6B). Examples of a few GO terms enriched are the cytoskeletal proteins (which correlates with the STRING and pathway analysis), GTP binding and GTPase activity and ATP binding and ATPase activity. 

The STRING analysis of the downregulated proteins revealed an interesting cluster of proteins: microtubule cytoskeleton cluster. This was of relevance to our study because we have found that fodrin executes a regulatory effect on microtubule nucleation [4]. We could also reveal the importance of α-fodrin in microtubule formation in the cells. The most important proteins that could be obtained as leads under this head are kinesin family members KIF2B, KIF3C and KIF23. Kinesins are microtubule-based motor proteins that move through them in a directed fashion carrying cargo in the cell. Kinesins are powered by the hydrolysis of ATP. Most kinesins move to the plus end of the microtubules with the exception of kinesin-14 family members [21]. KIF2B is a member of the kinesin-13 family. All reported members of this family are diffusive motors. As opposed to the other kinesins that display a stepwise regulated mechanical movement, these motors diffuse and often depolymerize the microtubule ends, in turn affecting microtubule dynamic instability [22]. It has been previously shown that KIF2B-deficient U2OS cells showed delayed movement of chromosomes towards the metaphase plate [23]. Through an independent project conducted in our lab, it was found that α-fodrin is important for the movement of chromosomes through the spindle during mitosis and subsequent arrangement at the metaphasic plate [5]. This finding correlates directly with the observation of reduced KIF2B upon α-fodrin downregulation. It is quite probable that the chromosome movement defects brought about by α-fodrin depletion may be through the downregulation of kinesins such as KIF2B. Another interesting kinesin that is downplayed in α-fodrin downregulated cells is KIF23, kinesin family member 23. KIF23 is a member of the kinesin-6 family. It is reported to be important in vesicular movement [24]. KIF23 is also key in antiparallel microtubule sliding. Although this role was conventionally assigned to the kinesin 5 family proteins, emerging reports in the field have shown multiple candidates such as KIF23 that can perform this function [25,26]. We have also found that α-fodrin depletion results in shorter spindles [5], which is probably due to the lack of sufficient antiparallel microtubule movement in the reduced presence of KIF23. KIF3C or kinesin family member 3C, important in cargo transport, was also found to be reduced in α-fodrin-depleted cells. This protein belongs to the kinesin super family 3 (KIF3). KIF3 proteins have been shown to be important for the axonal movement of vesicles rich in fodrin to help in neurite extension. Moreover, KIF3 deficiency also results in abnormal spindles [7,21]. These facts indicate that α-fodrin-based mitotic defects reported earlier by our group could occur through the regulation of these kinesins.

## 9. Conclusions

Global expression analysis of the proteome of α-fodrin downregulated U-251 MG cells revealed several interesting factors that are modulated to bring about the apparent effects of this molecule. Various regulatory pathways such as the cytoskeletal regulatory pathway, the apoptotic pathway and the cell cycle regulatory pathway were upregulated. This goes hand-in hand with our understanding of fodrin and also the literature produced by other groups. Our previous experiments showed α-fodrin’s regulation on the microtubule organization in cells (unpublished work and [5]). α-fodrin has also been shown to be important in apoptosis and cell cycle progression. We have also found that downregulation of α-fodrin significantly affects the microtubule cytoskeleton, notably, the spindle. In our study, we obtained shorter spindles in α-fodrin-depleted U-251 MG cells [5]. Independent work from our lab has also shown enrichment of abnormal spindles and chromosome congression defects in the absence of α-fodrin [5]. Through the proteome analysis, we found that kinesins such as KIF3C, KIF23 and KIF2B are downregulated, which could be the reason for the aforementioned spindle defects in these cells.

## Figures and Tables

**Figure 1 genes-12-00750-f001:**
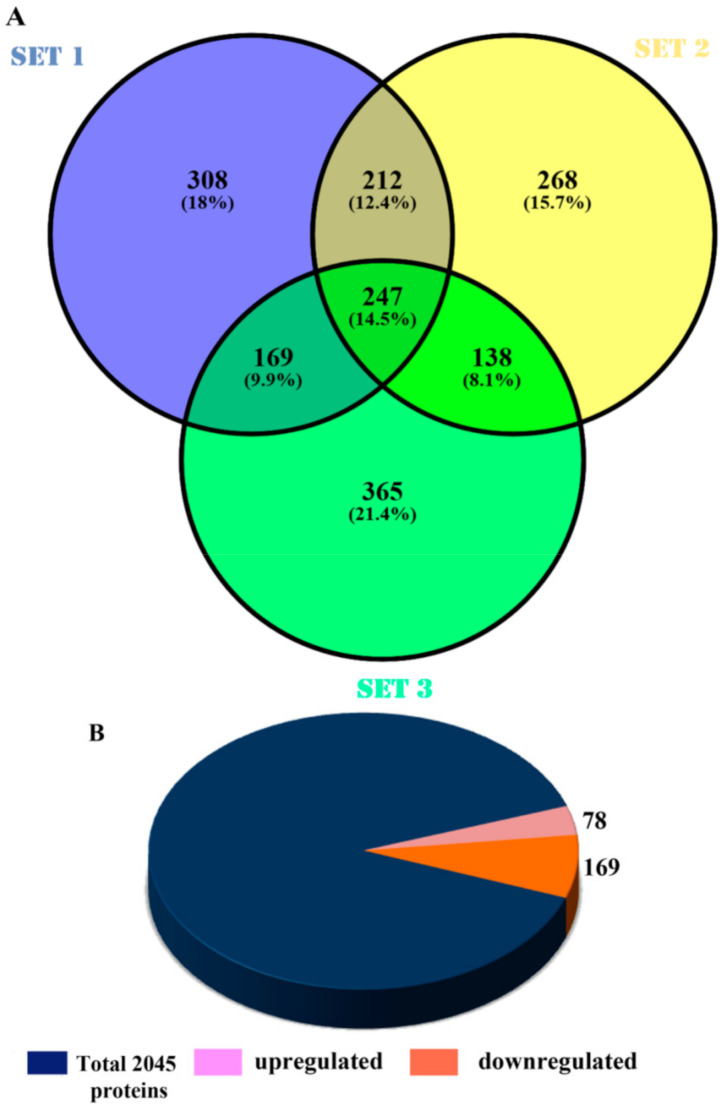
Results from VENNY 2.1. (**A**) Venn diagram showing the commonly affected proteins obtained in proteomic analysis. (**B**) Pie chart showing the fraction of the upregulated and downregulated proteins.

**Figure 2 genes-12-00750-f002:**
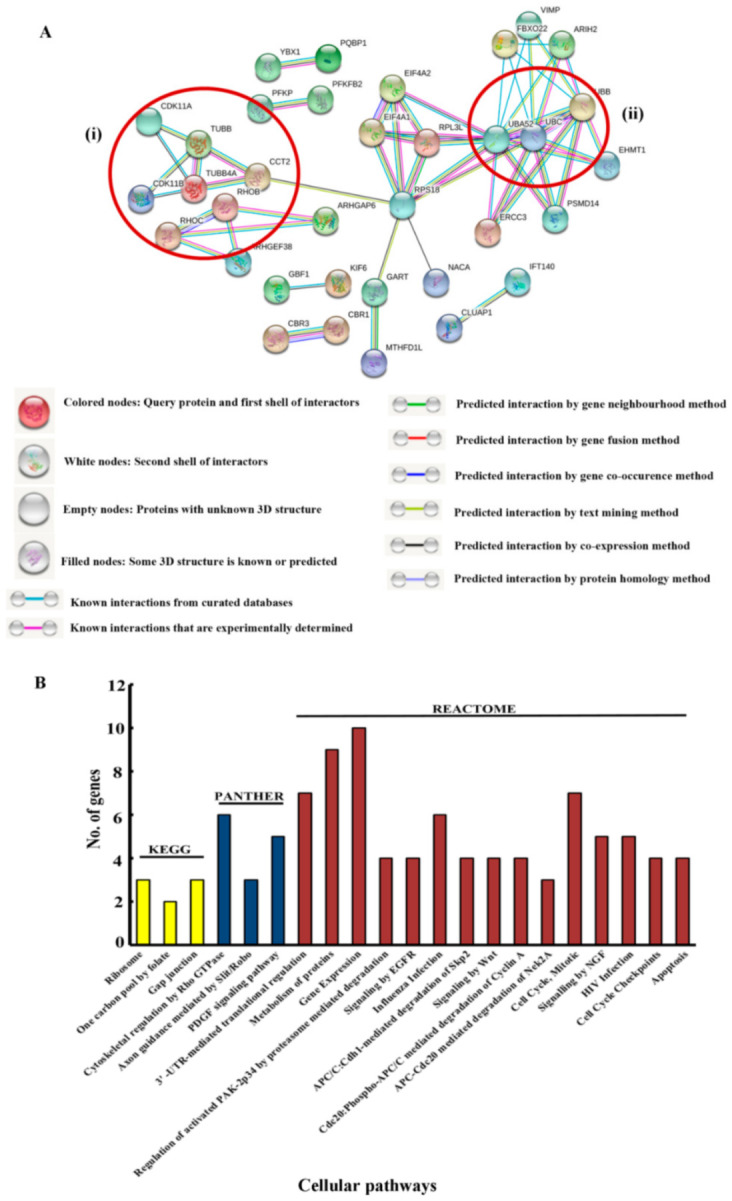
(**A**) The interactome of the upregulated proteins obtained from STRING v 11.0. The clusters corresponding to the (**i**) cytoskeletal regulatory proteins and cell cycle proteins and (**ii**) apoptotic proteins have been demarcated. Color coding of the nodes and networks is stated below the STRING output. (**B**) Gene Ontologies enriched by the upregulated proteins (pathways enriched as analyzed by DAVID).

**Figure 3 genes-12-00750-f003:**
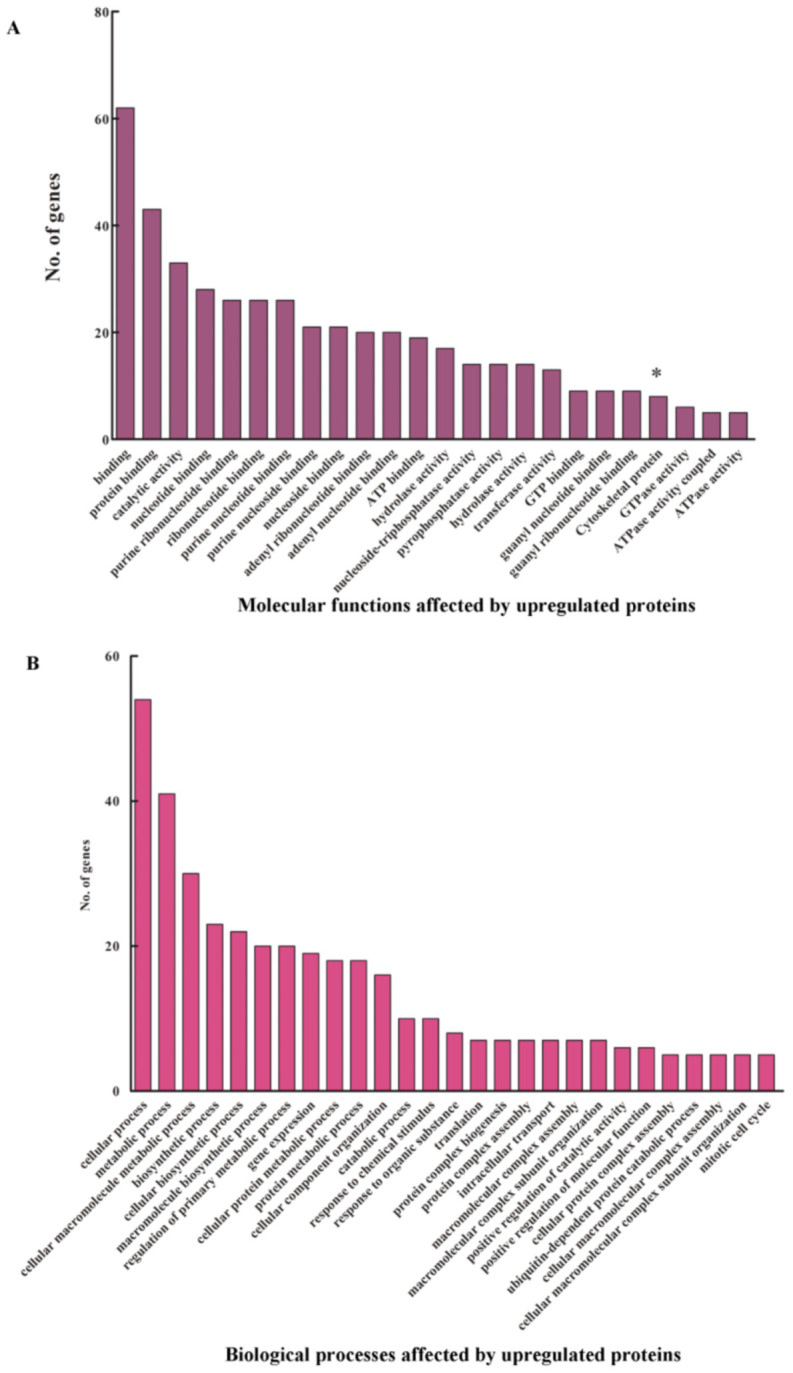
Analysis of the upregulated proteins. Gene Ontology terms. (**A**) Molecular functions and (**B**) biological processes enriched upon α-fodrin downregulation. * in (**A**) indicates genes enriched in cytoskeletal pathways.

**Figure 4 genes-12-00750-f004:**
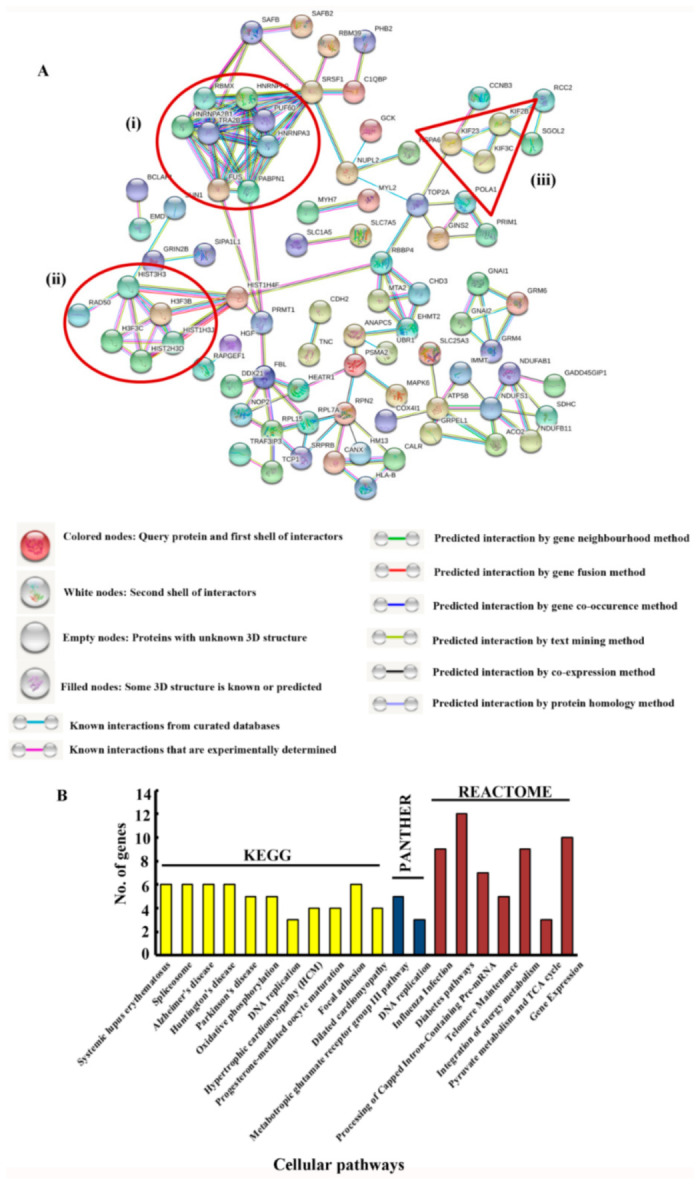
(**A**) The interactome of the downregulated proteins obtained from STRING v 11.0. The cluster corresponding to the (**i**) gene expression, (**ii**) DNA replication and repair and (**iii**) microtubule kinesins have been demarcated. Color coding of the nodes and networks is stated below the STRING output. (**B**) Gene Ontologies enriched by the upregulated proteins (pathways enriched as analyzed by DAVID).

**Figure 5 genes-12-00750-f005:**
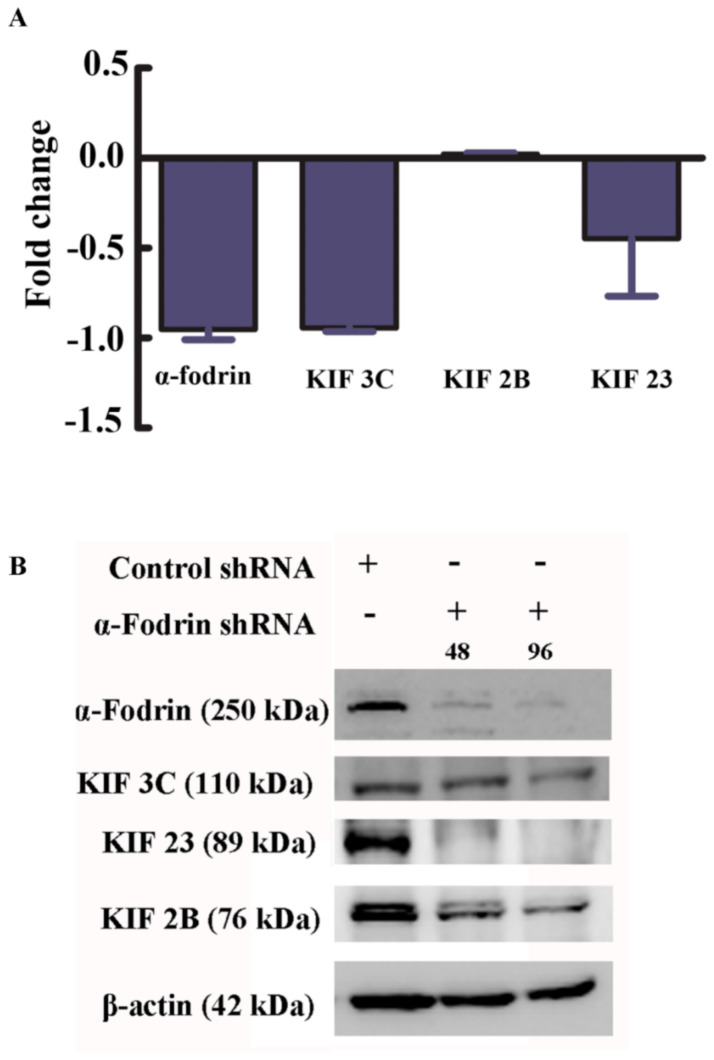
(**A**) Real-time qPCR validation of selected microtubule cytoskeleton-associated genes. Shown is the fold change expression of genes in α-fodrin shRNA-treated U-251 MG cells with respect to control shRNA-treated cells. β-actin is the internal control. (**B**) Western blot results showing the reduction in KIF 3C, KIF23 and KIF 2B upon α-fodrin downregulation post-α-fodrin shRNA treatment for 48 and 96 h. β-actin is used as loading control.

**Figure 6 genes-12-00750-f006:**
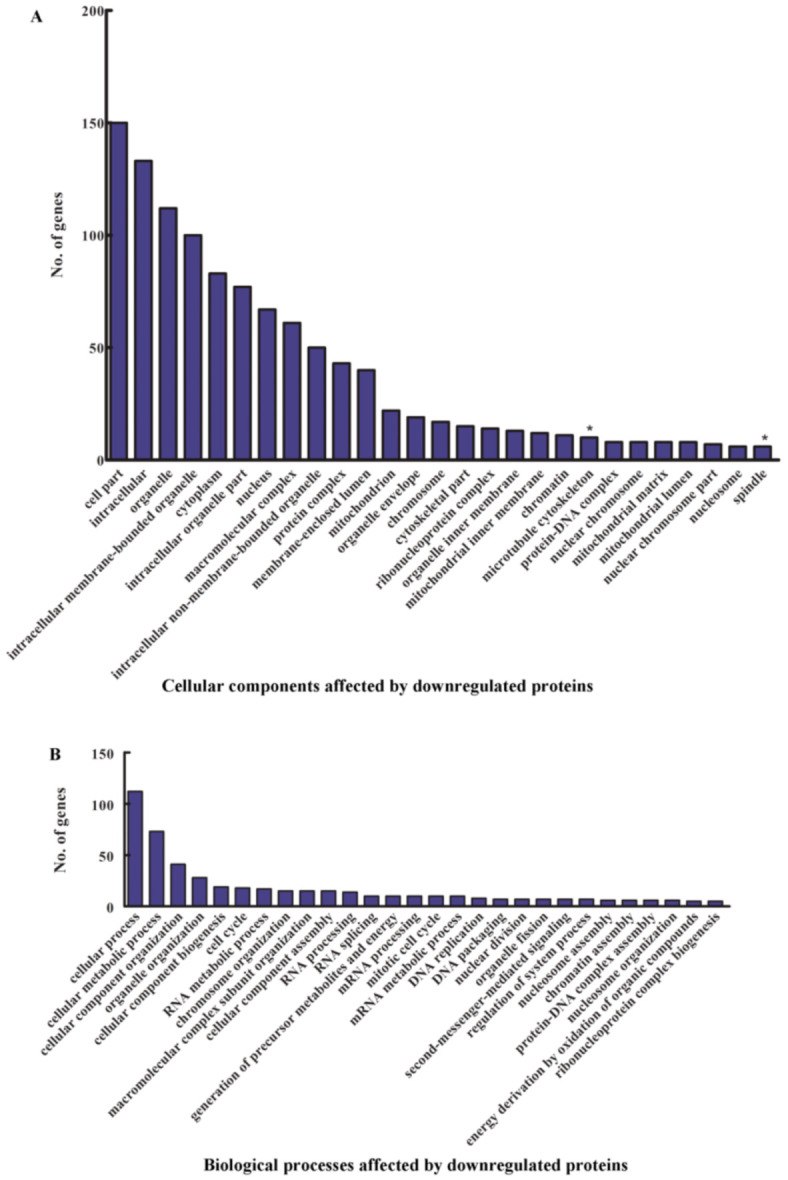
Analysis of the downregulated proteins. Gene Ontology terms. (**A**) Cellular components and (**B**) biological processes that were enriched upon the analysis of the downregulated proteins. * represents the microtubule cytoskeleton proteins and spindle-related proteins.

**Table 1 genes-12-00750-t001:** List of the forward and reverse primers used for the real-time quantitative PCR of the mentioned genes.

S. No.	Gene Name	Forward Primer (5′–3′)	Reverse Primer (5′–3′)
1.	*α-fodrin*	TCCCACCAACATCCAGCTTT	GCCTTGACAGCATCCTCACT
2.	*KIF2B*	AGCTGCAAGTCCTTGAGGATGG	TCCACCAGGTTCAGCACTTCCT
3.	*KIF3C*	TGGTGAGCGAGGAGAAGCAGAA	GGTGTGATCCATGATGTTCCTGC
4.	*KIF23*	GTAGCAAGACCTGTAGACAAGGC	TTCGCATGACGGCAAAGGTGGA
5.	*β-actin*	CACCATTGGCAATGAGCGGTTC	AGGTCTTTGCGGATGTCCACGT

## Data Availability

The mass spectrometry proteomics data have been deposited to the ProteomeXchange Consortium via the PRIDE [27] partner repository with the dataset identifier PXD023457 and 10.6019/PXD023457.

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
