# Peer review of "α-Fodrin in Cytoskeletal Organization and the Activity of Certain Key Microtubule Kinesins"

_genes, 2021, doi:10.3390/genes12050750_

Round 1

Reviewer 1 Report

Sreeja et al identified multiple potential genes, which up- or down regulated by depletion of a-fodrin by proteomic analysis with shRNA treated cells. This work found that a-fodrin might be involved in multiple pathways. Although authors provided useful datasets, the work significantly lacked validations of proteomic analysis and evidences for author’s conclusion.  Major concerns are listed below;

Concerns,

  1. Figure 1, There was no validations of shRNA used for this proteomic study. Authours must provide minimum of western blots (preferably as well as immunofluorescence) using multiple shRNAs.
  2. Related to Figure 1. To avoid effects of of-tartes of shRNAs used in proteomic data, authors should perform with multiple shRNA to validate datasets.
  3. Page 9, Where is Table 3?
  4. Page 9, In addition to evaluate mRNA levels, validations in protein expressions are required. Especially, authors could not validate consistency between proteomic data and mRNA expression in Kif2B. Further validations in both up-regulated and down-regulated candidates from multiple pathways authors identified are required.
  5. Authors found that a-fodrin may be involved in multiple pathway; however, unfortunately, this study completely lack the mechanism of how a-fodrin regulates those pathways.
  6. It is unclear how the up-regulated or down-regulated genes by a-fodrin are impacted functions in viability, cell cycle, cytoskeleton, and more.
  7. Page 14, Authors indicated that authors found a depletion of a-fodrin causes mitotic defects, that might relate the potential down regulation of Kif2B and/or Kif3C. Why don’t authors test directly the hypothesis, rather than speculation?
  8. Page 13, citation format “Ishikawa et al., 1983
  9. Page 13, Authors should separate results and discussion. This page included many speculations based on other studies.
  10. Abstract, “…in vitro and in vivo approaches…’ is overstatement.

Author Response

Reviewer 1:

We thank the reviewer for his suggestions. However, it has to be pointed out that the presented proteomic study is comprehensive and intends to provide with an overall understanding of the functionality of fodrin. In the interest of the researchers in this field and also for the larger scientific community, we have submitted all the data sets pertaining to the study in the Protein exchange consortium. Upon publication of this manuscript the datasets will be made accessible. Hence, interested researchers can dive deep into the data and do an in-depth analysis in accordance with their research focus.  

  1. Figure 1, There was no validations of shRNA used for this proteomic study. Authors must provide minimum of western blots (preferably as well as immunofluorescence) using multiple shRNAs.

We would like to direct the reviewer to our earlier publication Nellika et al., Cell cycle 2019, where we have validated the depletion of α-fodrin using multiple shRNAs. In this report as well we have added the western blot analysis to show the reduction of α-fodrin (Figure 5B). 

  1. Related to Figure 1. To avoid effects off-target of shRNAs used in proteomic data, authors should perform with multiple shRNA to validate datasets.

The shRNAs used in the study are a cocktail of sequences binding to various parts of the α-fodrin gene to ensure optimal downregulation. However performing a whole proteomic analysis using a different set of shRNAs is beyond the scope of this study.

  1. Page 9, Where is Table 3?

Table 3 is given separately as supplementary table 2.

  1. Page 9, In addition to evaluate mRNA levels, validations in protein expressions are required. Especially, authors could not validate consistency between proteomic data and mRNA expression in Kif2B. Further validations in both up-regulated and down-regulated candidates from multiple pathways authors identified are required.

As the reviewer has rightly pointed out, we could not validate the reduction in the mRNA levels of KIF2B, hence we have added a western blotting figure (Figure. 5B) where we are showing the reduction of KIF2B, KIF3C and KIF23. Such validation of all the 247 up-regulated and downregulated proteins is beyond the scope of this work.

  1. Authors found that α-fodrin may be involved in multiple pathway; however, unfortunately, this study completely lack the mechanism of how α-fodrin regulates those pathways.

α-Fodrin is involved in multiple cellular pathways. Through the leads that we could obtain in our proteomic analysis, we have tried to bring into context the various reported functions of α-fodrin. We have discussed this at length. However, as earlier stated, the aim of this study is to give a peripheral perspective and not to do a profound analysis.

  1. It is unclear how the up-regulated or down-regulated genes by α-fodrin are impacted functions in viability, cell cycle, cytoskeleton, and more.

This has been discussed at length in the discussion.

  1. Page 14, Authors indicated that authors found a depletion of α-fodrin causes mitotic defects, that might relate the potential down regulation of Kif2B and/or Kif3C. Why don’t authors test directly the hypothesis, rather than speculation?

This a very promising idea and our laboratory will be working towards it. However, it is understandable that evaluating the effect of certain kinesins in mitosis would be a separate study in itself and cannot be clubbed with the present work.

  1. Page 13, citation format “Ishikawa et al., 1983

The citation has been changed in the expected format.

  1. Page 13, Authors should separate results and discussion. This page included many speculations based on other studies.

The results and discussion are separate. However, there could be possible overlap to a certain degree to maintain continuity and flow.

10. Abstract, “…in vitro and in vivo approaches…’ is overstatement.

This statement has been appropriately rectified to “Our earlier studies involving in vitro and ex vivo approaches have led us to identify certain undiscovered characteristics of α-fodrin, a prominent cortical protein.”

Reviewer 2 Report

In this study, authors using the label-free LC-MS/MS method have investigated the role of α-fodrin in U-251 MG glioblastoma cells. The approach in this study was to downregulate the expression of α-fodrin using shRNA and to quantify global proteomic profiles.

Overall, the manuscript is well-drafted, with an excellent discussion of results. However, I would like the authors to clarify/rectify these points in the next version.

1) Why was glioblastoma cells U-251 MG chosen in this study?

2) Was FDR correction performed for the ANOVA?

3) In the methods for the identification of commonly affected proteins, the authors mention that they used VENNY to identify 247 proteins.  From the methods and Figure 1 A, I understand that 247 proteins were common across the 3 sets. I would like to know if they found all these 247 genes DE in the same direction in all three sets.

4) Figure 3A. Specify in the legend what the Astrix indicates.

5) Figure 6A: correct the title to ‘Cellular component affected by downregulated proteins.’

6) Authors state in the discussion that ‘Fodrin has also been shown to re-polarize under conditions of stress in specific cells such as chromaffin cell’. Please explain and add a reference to this statement.

Author Response

Reviewer 2:

We thank the reviewer for the appreciation of our work and for raising pertinent questions. We have addressed them below.

1) Why was glioblastoma cells U-251 MG chosen in this study?

This is a valid concern and we have addressed this in para 3 of the Introduction.

2) Was FDR correction performed for the ANOVA?

Yes FDR correction was done for the ANOVA analysis.

3) In the methods for the identification of commonly affected proteins, the authors mention that they used VENNY to identify 247 proteins.  From the methods and Figure 1 A, I understand that 247 proteins were common across the 3 sets. I would like to know if they found all these 247 genes DE in the same direction in all three sets.

Out of the 247 proteins 78 were upregulated and 169 were downregulated (as given in Results, Protein identification)

4) Figure 3A. Specify in the legend what the Astrix indicates.

The following sentence has been added in the legend of Figure. 3 “* in A indicates genes enriched in cytoskeletal pathways”.

5) Figure 6A: correct the title to ‘Cellular component affected by downregulated proteins.’

Thank you for pointing this out. We have made the necessary correction in the figure title.

6) Authors state in the discussion that ‘Fodrin has also been shown to re-polarize under conditions of stress in specific cells such as chromaffin cell’. Please explain and add a reference to this statement.

Appropriate citations have been added.

Reviewer 3 Report

In this manuscript Sreeja et al. address the expanding framework of known functions for fodrin, a central component of the cortical cytoskeleton. The cortical cytoskeleton is a complex structure that is important for many cellular functions. It comprises a myriad of interactions involving actin and microtubules and consists of more general as well as highly specialized connections mediating specific functional and special processes. The spectrin family are multifaceted proteins that facilitate connections between different aspects of the cortical cytoskeleton and, thus, are also important for an expanding number of known functions. Among results emerging from other studies, the Sengupta lab has recently shown alpha-fodrin is important for regulating / coordinating gamma-tubulin nucleation activity, and for chromosome congression and successful mitosis. In the current study, Sreeja et al. undertake a global proteomics analysis to identify proteins whose levels are impacted in an alpha-fodrin-dependent manner. The results reveal that the absence of alpha-fodrin results in upregulation and downregulation of a specific set of proteins. Analysis of these proteins revealed interesting connections with multiple major cellular pathways, which further connects fodrin to known functions and implicates it in a range of potential roles. Moreover, the authors go on to validate changes in the level of several kinesin proteins identified as hits in their global analysis. The results presented expand on the knowledge of this far-reaching protein family that is important beyond its best-known role in the cortical cytoskeleton. The results also set the stage for what will likely be more discoveries in the expanding range of fodrin functions. There are a number of issues with the current manuscript that should be addressed.

Issues:

In Figure 3 the legend is very short and not very descriptive. More info is needed. In the results it is stated that the authors found 78 upregulated proteins and 169 downregulated. At least this is the number that passes the cutoff criteria. Figure 3 legend states simply “Analysis of upregulated proteins” but there are perhaps 400 genes in total represented by the bars/categories in Fig 3B, and a high number in 3A (not 78). More explanation is needed here. Are individual genes counted in multiple categories? Should that be the case? If so, the authors should address this aspect of the analysis and how it may impact the identification of enriched categories.

The authors suggest that they may not have observed differences in KIF2B mRNA levels due to a relatively longer half-life. This may be true, but also leaves open the possibility that KIF2B may be a false positive. Could the levels of KIF2B be quantified by western blot or immunoflorescence in the alpha-fodrin knockdown cells?   As discussed in the text, KIF2B is a tantalizing hit because of its role in chromosome movement and previous work from the Sengupta lab showing loss of fodrin inhibits successful chromosome congression in metaphase. It seems unfortunate to leave these KIF2B results unclarified with regard to potential half-life issues.  Can alternative methods be used to quantify KIF2B protein levels and/or measure KIF2B mRNA half-life?

Additionally, the choice of actin as a housekeeping gene that will have consistent levels following alpha-fodrin knockdown does not seem obvious, considering the enrichment of cytoskeletal proteins with altered expression levels. Also, would one predict a more than 1 fold decrease in alpha-fodrin mRNA levels following efficient shRNA treatment? Can the actin level be verified before and after alpha-fodrin knockdown?

The discussion of kinesin proteins must be corrected. First, as stated “Most kinesins move to the plus end of the microtubules with the exception of kinesin-14 family members.”  This should really make clear then that kinesin-14 motors move toward the minus end. This is not obvious because it was critically left out that Kinesin-13 motors (such as KIF2B) do not move directly toward the plus or minus end. They instead diffuse on microtubules. They don’t ‘walk’ like other kinesins, but they instead depolymerize both the plus and minus ends if they encounter the ends. This is a critical omission considering that KIF2B may be a hit of potential importance.

The authors do not include kinein-5 family in their discussion. These are the main anti-parallel microtubule crosslinking kinesins in the mitotic spindle. They previously found that spindles in cells lacking fodrin are shorter and speculate that may be due to reduced levels of KIF23, which has been seen to crosslink antiparallel microtubules in vitro. The authors should make clear whether there is existing evidence that KIF23 crosslinks microtubules in spindles (with citation) or whether they are speculating, based on their data, that KIF23 may play some as yet unrecognized role in crosslinking microtubules in the spindle.

Figure 4 legend states it is upregulated proteins. According to the text, these are downregulated proteins. There may be additional confusion between the text and legend. At the top of page 9 it reads as if the gene expression, microtubule cytoskeleton proteins and DNA replication protein clusters are highlighted. In the legend it states cytoskeletal regulatory proteins, cell cycle proteins, and apoptotic protein clusters have been demarcated.  Which are highlighted and why are they the more interesting?  Also, the demarcations should be labeled with their corresponding clusters.

In Figure 6 there is the same issue as figure 3. The legend is very short and not descriptive. In the results it is stated that the authors found 169 downregulated proteins. This number is essentially shown in each of the first few categories alone in panel A. How do the authors (and readers) interpret this? More explanation is needed here. Surely individual genes are counted in multiple categories. The authors should address this aspect of the analysis and how it may impact the identification of enriched categories.

In figure 6, the authors highlight the microtubule cytoskeleton and spindle categories. The spindle category contains the least genes. Is this number enriched compared with all spindle proteins?

The authors have submitted the mass spec results to a public database. The results would likely be more accessible to readers if the hits above cut-off were included as a supplemental table or excel sheet.

Minor Issues:

Why did the authors select the three kinesin proteins for validation? More insight into the relevance and potential importance would be helpful.

In the discussion the authors state at the end of the second paragraph that “it is indicated that absence of fodrin may indirectly affect the dynamic instability at the plus ends.” Although this is an accurate statement. Additional context may be helpful. For instance, are there specific biological instances (healthy or diseased cells) that lack fodrin and may suffer this effect? Alternatively, could one also speculate that fodrin may normally have some, perhaps poorly defined, effect on plus end dynamics? It may be nice to include discussion about the potential roles of fodrin in addition to its absence.

Page 9, “This was of relevance to our study because we have found the regulating effect of fodrin on microtubule nucleation. We could also reveal the importance of alpha-fodrin in microtubule formation in the cells.”  Both of these statements need to be supported with appropriate citations. Also, the second sentence is written as a possibility. It is unclear if the authors are saying that they have revealed it in published data, have found it in unpublished data, or could possibly find it with future experiments. This should be clarified.

The legend for Fig 2A states that three clusters have been demarcated. But only two are circled in the network. Why are they circled? Also, it is not immediately clear which clusters correspond to which proteins. Can this be labeled?

End of paragraph on page 7. “The cluster obtained in the STRING output also showed up in the pathway analysis. Is there 3 or only 1 cluster? If 3, then the sentence should read “The cytoskeletal protein cluster obtained in the STRING output…”.  Also, the end states (Figure 3A and B) (marked with * in the graph).  However, a “*” only appears in Figure 3A.

Providing examples of enriched GO terms in the discussion is potentially interesting, but this type of presentation would perhaps benefit from some discussion of the potential implications. At the minimum, a touch of discussion would be helpful indicating whether the authors are just pointing out some terms? Or are they suggesting that fodrin function normally impacts these areas/processes?

In Fig 2A it is hard to tell which interactions contain lines of which colors. If any codes are not needed can they be removed from the key (e.g. the white nodes may not exist in the displayed network?)

Statements such as “However, it has been reported that actin reorganizes under conditions of stress to form actin bundles.” in the discussion need to have citations. The following sentence also. If these all belong with the citation [9] at the end of the paragraph this is unclear and perhaps the citation should be repeated to support specific statements.

Figure 6 graphs state downregulated proteins but yet the legend states upregulated proteins.

“Quantitative Real Time PCR” subheading possibly should be italics and non-bold.

In abstract correct typo, “…have led us to identify certain undiscovered characteristics of …”

In methods for protein extraction, 50 Mm ammonium bicarbonate should read 50 mM, and the phrase “protease inhibitor” does not need to be capitalized.

Author Response

Reviewer 3:

We thank the reviewer 3 for very suitable comments. Specific response to the comments are,

  1. 1. In Figure 3 the legend is very short and not very descriptive. More info is needed. In the results it is stated that the authors found 78 upregulated proteins and 169 downregulated. At least this is the number that passes the cut-off criteria. Figure 3 legend states simply “Analysis of upregulated proteins” but there are perhaps 400 genes in total represented by the bars/categories in Fig 3B, and a high number in 3A (not 78). More explanation is needed here. Are individual genes counted in multiple categories? Should that be the case? If so, the authors should address this aspect of the analysis and how it may impact the identification of enriched categories.

Figure 3 legend has been modified accordingly. An explanation regarding the overlapping of proteins in multiple GO terms has been given in the “Protein identification” section of the Results.

“Various protein databases were used to analyse these proteins. Since cellular pathways are highly interconnected and do not work in isolation, there is considerable overlap and hence different proteins can occur in multiple pathways and this accounts for the redundancy in the presented data. However, it should be noted that this does not affect our interpretation because only significantly enriched clusters have been detailed.”

  1. The authors suggest that they may not have observed differences in KIF2B mRNA levels due to a relatively longer half-life. This may be true, but also leaves open the possibility that KIF2B may be a false positive. Could the levels of KIF2B be quantified by western blot or immunoflorescence in the alpha-fodrin knockdown cells?   As discussed in the text, KIF2B is a tantalizing hit because of its role in chromosome movement and previous work from the Sengupta lab showing loss of fodrin inhibits successful chromosome congression in metaphase. It seems unfortunate to leave these KIF2B results unclarified with regard to potential half-life issues.  Can alternative methods be used to quantify KIF2B protein levels and/or measure KIF2B mRNA half-life?

To address this concern we have added a western blotting image (Figure. 5B) to validate the reduction in the expression of KIF 2B, KIF 3C and KIF 23. A sentence referring to the figure has been added in 2nd para of the section “Analysis of downregulated proteins”. The legend of Figure. 5 has been modified accordingly.

  1. Additionally, the choice of actin as a housekeeping gene that will have consistent levels following alpha-fodrin knockdown does not seem obvious, considering the enrichment of cytoskeletal proteins with altered expression levels. Also, would one predict a more than 1 fold decrease in alpha-fodrin mRNA levels following efficient shRNA treatment? Can the actin level be verified before and after alpha-fodrin knockdown?

As you can appreciate from our data (Figure 5B) and also from our previously published work (Nellika et al.), we have shown that the expression levels of housekeeping genes such as β-actin and α-tubulin do not alter upon α-fodrin depletion. In Nellika et al., Figure 5C, we verified the mRNA level change in α-fodrin using our shRNA and found upto 68% reduction. Reduction in α-fodrin expression level has also been seen in immunofluorescence experiments (unpublished).   

  1. The discussion of kinesin proteins must be corrected. First, as stated “Most kinesins move to the plus end of the microtubules with the exception of kinesin-14 family members.”  This should really make clear then that kinesin-14 motors move toward the minus end. This is not obvious because it was critically left out that Kinesin-13 motors (such as KIF2B) do not move directly toward the plus or minus end. They instead diffuse on microtubules. They don’t ‘walk’ like other kinesins, but they instead depolymerize both the plus and minus ends if they encounter the ends. This is a critical omission considering that KIF2B may be a hit of potential importance.

This concern is valid and we thank the reviewer for raising it. The following explanation (with citation) regarding the same has been included in 6th para of Discussion.

“KIF2B is a member of the kinesin-13 family. All reported members of this family are diffusive motors. As opposed to the other kinesins that display a stepwise regulated mechanical movement, these motors diffuse and often depolymerize the microtubule ends in turn affecting microtubule dynamic instability [22].”

  1. The authors do not include kinein-5 family in their discussion. These are the main anti-parallel microtubule crosslinking kinesins in the mitotic spindle. They previously found that spindles in cells lacking fodrin are shorter and speculate that may be due to reduced levels of KIF23, which has been seen to crosslink antiparallel microtubules in vitro. The authors should make clear whether there is existing evidence that KIF23 crosslinks microtubules in spindles (with citation) or whether they are speculating, based on their data, that KIF23 may play some as yet unrecognized role in crosslinking microtubules in the spindle.

As rightly pointed out by the reviewer, we have included the following explanation with appropriate citations in para 6th of the Discussion.

“KIF23 is a member of the kinesin-6 family. It is reported to be important in vesicular movement [24]. KIF23 is also key in antiparallel microtubule sliding. Although this role was conventionally assigned to the kinesin 5 family proteins but emerging reports in the field have shown multiple candidates such as KIF23 that can perform this function [25,26]. We have also found that α-fodrin depletion results in shorter spindles [5] which probably is because of the lack of sufficient antiparallel microtubule movement in the reduced presence of KIF23.”

  1. Figure 4 legend states it is upregulated proteins. According to the text, these are downregulated proteins. There may be additional confusion between the text and legend. At the top of page 9 it reads as if the gene expression, microtubule cytoskeleton proteins and DNA replication protein clusters are highlighted. In the legend it states cytoskeletal regulatory proteins, cell cycle proteins, and apoptotic protein clusters have been demarcated.  Which are highlighted and why are they the more interesting?  Also, the demarcations should be labeled with their corresponding clusters.

Of the 78 upregulated and 169 downregulated proteins we have selected a few for discussion considering the research focus of our lab and also the general interest of the readers. The following statement to this effect has been added in the Protein identification part of Results.

“In the present study, we have also discussed in detail a few interesting proteins in the upregulated and the downregulated cohort. These proteins are of general interest and also overlap with the research focus of our laboratory.” 

Figure 2 and 4 and their corresponding legends have been modified to clarify the details of the enriched clusters.

  1. In Figure 6 there is the same issue as figure 3. The legend is very short and not descriptive. In the results it is stated that the authors found 169 downregulated proteins. This number is essentially shown in each of the first few categories alone in panel A. How do the authors (and readers) interpret this? More explanation is needed here. Surely individual genes are counted in multiple categories. The authors should address this aspect of the analysis and how it may impact the identification of enriched categories.

The issue has been addressed in response to the 1st comment. The legend of figure. 6 has been modified accordingly.

  1. In figure 6, the authors highlight the microtubule cytoskeleton and spindle categories. The spindle category contains the least number of genes. Is this number enriched compared with all spindle proteins?

As mentioned in the Methods section, the GO categories obtained as hits from pathway analysis are generated after they pass an Expression Analysis Systematic Explorer (EASE) score (a modified Fisher’s Exact P-value) of ≤0.1  for each term. Hence all the pathways shown in the graph are significant. The number is enriched as compared to all spindle proteins.

  1. The authors have submitted the mass spec results to a public database. The results would likely be more accessible to readers if the hits above cut-off were included as a supplemental table or excel sheet.

A detailed tabulated form of the downregulated and upregulated proteins has been added in the manuscript as separate supplemental tables.

Minor Issues:

  1. Why did the authors select the three kinesin proteins for validation? More insight into the relevance and potential importance would be helpful.

The following explanation regarding the raised concern has been added in the 2nd para of the section titled “Analysis of downregulated proteins”.

“Literature survey of these kinesins (as will be detailed later) showed their relevance in various processes such as chromosome congression, spindle morphology and cargo transport. These pathways resonated with our understanding of fodrin functions. Hence it was imperative for us to analyse and validate them.”  

  1. In the discussion the authors state at the end of the second paragraph that “it is indicated that absence of fodrin may indirectly affect the dynamic instability at the plus ends.” Although this is an accurate statement. Additional context may be helpful. For instance, are there specific biological instances (healthy or diseased cells) that lack fodrin and may suffer this effect? Alternatively, could one also speculate that fodrin may normally have some, perhaps poorly defined, effect on plus end dynamics? It may be nice to include discussion about the potential roles of fodrin in addition to its absence.

To address the involvement of fodrin in microtubule dynamic instability the following explanation has been included in the 2nd para of the Discussion.

“Independent of this, it was also found that stathmin was downregulated upon α-fodrin depletion (Table: 3). Stathmin is a microtubule depolymerising protein. Reduction of the same via activation of Rho GTPases can thus result in stable microtubules. Fodrin has been shown to associate with tubulin and induce microtubule bundling in vitro [14]. However, there is no direct evidence to validate the effect of fodrin on microtubule plus end dynamics. Thus through the presented proteomic evidence, it is indicated that absence of α-fodrin may indirectly affect the dynamic instability of microtubules.”  

  1. Page 9, “This was of relevance to our study because we have found the regulating effect of fodrin on microtubule nucleation. We could also reveal the importance of alpha-fodrin in microtubule formation in the cells.”  Both of these statements need to be supported with appropriate citations. Also, the second sentence is written as a possibility. It is unclear if the authors are saying that they have revealed it in published data, have found it in unpublished data, or could possibly find it with future experiments. This should be clarified.

The statements have been modified and appropriate citations have been added in the 2nd para of the “Analysis of the downregulated proteins” section.

“This was of relevance to our study because we have earlier described the regulatory effect of fodrin on microtubule nucleation [4]. We have also understood the importance of α-fodrin in microtubule formation in the cells (unpublished data).

  1. The legend for Fig 2A states that three clusters have been demarcated. But only two are circled in the network. Why are they circled? Also, it is not immediately clear which clusters correspond to which proteins. Can this be labeled?

These have been demarcated because they were interesting and relevant to the research area of our group. Hence they have been discussed in detail and the corresponding clusters in the figure have been labelled.

  1. End of paragraph on page 7. “The cluster obtained in the STRING output also showed up in the pathway analysis. Is there 3 or only 1 cluster? If 3, then the sentence should read “The cytoskeletal protein cluster obtained in the STRING output…”.  Also, the end states (Figure 3A and B) (marked with * in the graph).  However, a “*” only appears in Figure 3A.

The segment has been modified as follows

“The cytoskeletal protein cluster obtained in the STRING output also showed up in the pathway analysis (Figure. 3 A, marked with * in the graph). Biological processes such as gene expression, translation and intracellular transport are also affected by the upregulated proteins (Figure. 3B).”

  1. Providing examples of enriched GO terms in the discussion is potentially interesting, but this type of presentation would perhaps benefit from some discussion of the potential implications. At the minimum, a touch of discussion would be helpful indicating whether the authors are just pointing out some terms? Or are they suggesting that fodrin function normally impacts these areas/processes?

Fodrin is a massive protein with binding domains for various interacting partners. By virtue of this, it is important in many cellular pathways. Through the leads obtained in the proteomic analysis, we have tried to bring in focus, the various potential functions of fodrin. This we have now discussed at length in our manuscript.

  1. In Fig 2A it is hard to tell which interactions contain lines of which colors. If any codes are not needed can they be removed from the key (e.g. the white nodes may not exist in the displayed network?)

All colour codes and interactions are required and cannot be removed.

  1. Statements such as “However, it has been reported that actin reorganizes under conditions of stress to form actin bundles.” in the discussion need to have citations. The following sentence also. If these all belong with the citation [9] at the end of the paragraph this is unclear and perhaps the citation should be repeated to support specific statements.

Appropriate citations has been included in the 1st para of the discussion.

  1. Figure 6 graphs state downregulated proteins but yet the legend states upregulated proteins.

We thank the reviewer for pointing this. The legend of the Figure. 6 has been modified to address this concern.

  1. “Quantitative Real Time PCR” subheading possibly should be italics and non-bold.

This subheading has italicized and maintained in bold.

  1. In abstract correct typo, “…have led us to identify certain undiscovered characteristics of …”

The abstract has been changed accordingly.

  1. In methods for protein extraction, 50 Mm ammonium bicarbonate should read 50 mM, and the phrase “protease inhibitor” does not need to be capitalized.

 These corrections have been made.
